# Differential Effects of the Flavonolignans Silybin, Silychristin and 2,3-Dehydrosilybin on *Mesocestoides vogae* Larvae (Cestoda) under Hypoxic and Aerobic In Vitro Conditions

**DOI:** 10.3390/molecules23112999

**Published:** 2018-11-16

**Authors:** Gabriela Hrčková, Terézia Mačák Kubašková, Oldřich Benada, Olga Kofroňová, Lenka Tumová, David Biedermann

**Affiliations:** 1Department of Experimental Pharmacology, Institute of Parasitology, Slovak Academy of Sciences, Hlinkova 3, SK 040 01 Košice, Slovakia; kubaskova@saske.sk; 2Laboratory of Molecular Structure Characterization, Institute of Microbiology of the Czech Academy of Sciences, Vídeňská 1083, CZ 142 20 Prague, Czech Republic; benada@biomed.cas.cz (O.B.); kofra@biomed.cas.cz (O.K.); 3Department of Pharmacognosy, Faculty of Pharmacy, Charles University, Heyrovského 1203, CZ 501 65 Hradec Králové, Czech Republic; tumova@faf.cuni.cz; 4Laboratory of Biotransformation, Institute of Microbiology of the Czech Academy of Sciences, Vídeňská 1083, CZ 142 20 Prague, Czech Republic

**Keywords:** silybin, 2,3-dehydrosilybin, silychristin, *Mesocestoides vogae* larvae, aerobic and hypoxic cultivation

## Abstract

*Mesocestoides vogae* larvae represent a suitable model for evaluating the larvicidal potential of various compounds. In this study we investigated the in vitro effects of three natural flavonolignans—silybin (SB), 2,3-dehydrosilybin (DHSB) and silychristin (SCH)—on *M. vogae* larvae at concentrations of 5 and 50 μM under aerobic and hypoxic conditions for 72 h. With both kinds of treatment, the viability and motility of larvae remained unchanged, metabolic activity, neutral red uptake and concentrations of neutral lipids were reduced, in contrast with a significantly elevated glucose content. Incubation conditions modified the effects of individual FLs depending on their concentration. Under both sets of conditions, SB and SCH suppressed metabolic activity, the concentration of glucose, lipids and partially motility more at 50 μM, but neutral red uptake was elevated. DHSB exerted larvicidal activity and affected motility and neutral lipid concentrations differently depending on the cultivation conditions, whereas it decreased glucose concentration. DHSB at the 50 μM concentration caused irreversible morphological alterations along with damage to the microvillus surface of larvae, which was accompanied by unregulated neutral red uptake. In conclusion, SB and SCH suppressed mitochondrial functions and energy stores, inducing a physiological misbalance, whereas DHSB exhibited a direct larvicidal effect due to damage to the tegument and complete disruption of larval physiology and metabolism.

## 1. Introduction

Natural products are an important source of structural motifs in drug discovery. About half of the small molecules approved as new drugs by FDA each year come directly, or are semi-synthesized, from natural products [1]. Secondary metabolites produced by the higher plants are a very rich source of molecules with anthelmintic activity [2] among which flavonoids/polyphenols, essential oils, condensed tannins and sesquiterpene lactones exhibit the most promising results [3,4,5]. There are very few studies that have examined the role of flavonoids and polyphenols against flatworm parasites. For example, the flavonoid genistein has been shown to have anthelmintic activity on the cestode *Raillietina echinobothrida* in vitro by causing flaccid paralysis accompanied by alterations in tegumental architecture [6,7]. Metacestocidal activities of genistein and other synthetic isoflavones were demonstrated in vitro against *Echinococcus multilocullaris* and *Echinococcus granulosus* [8]. The polyphenol curcumin was shown to cause the death of all *Schistosoma mansoni* worms in vitro at concentrations of 50 and 100 μM. All pairs of coupled adult worms were separated by the action of curcumin at doses of 20–100 μM, and it also reduced egg production by 50% [9].

Silymarin is a mixture of phenolic molecules isolated from the fruits of *Silybum marianum* L. (Asteraceae) and the major components (about or more than 10% relative abundance in HPLC) are the flavonolignans silybin A, silybin B, silychristin A, silydianin and isosilybin A (Figure 1) *Silybum marianum* has been used for centuries in the treatment of liver diseases (first mentioned in Hieronimus Bock’s *Kreuterbuch*, 1546) and the major biological activities of silymarin and silybin include antioxidant/prooxidant, hepatoprotective and chemoprotective (for a review, see [10]). In addition, silymarin/silybin exhibits immunomodulatory effects with both immunostimulatory and immunosuppressive activities [11]. Among other beneficial activities of silymarin/silybin that are not directly linked to antioxidant effects (radical scavenging), is an anti-cancer effect shown both in vitro and in vivo, as well as hypocholesterolemic, cardioprotective, antidiabetic and neuroprotective activities, which indicate a pleiotropic mode of action of its flavonolignans [12,13]. However, much less is known about the biological activities of other flavonolignans and their derivatives. In recent years, scientific interest has been focused on 2,3-dehydrosilybin and other dehydro-compounds, revealing interesting biological activities [14,15,16]. So far, the antiparasitic potential of individual silymarin flavonolignans has not been considered, although during infections with *Schistosoma mansoni*, silymarin alone and even more in adjunct therapy with praziquatel, caused a significant decrease in worm burdens and hepatic tissue egg load with an increase in the percentage of dead eggs. In addition, combined therapy alleviated liver inflammation and fibrosis [17]. Fibrosis is also a hallmark of many cestode infections, including alveolar echinococcosis and experimental infection with the tetrathyridium of *Mesocestoides vogae* (syn. *M. corti*). It is considered to be a suitable laboratory model for studying cestode biology, host-parasite interactions and the antiparasitic potential of various compounds. In our previous studies on mice with *M. vogae* infection, silymarin in combination with praziquantel, an increased efficacy of the drug praziquantel compared to monotherapy was found, along with a decrease in liver fibrosis and liver damage due to oxidative stress [18,19]. In this study, we investigated the spectrum of activities of selected flavonolignans (FLs) such as silybin, silychristin and dehydrosilybin on this larval cestode model under in vitro conditions. It was shown that the tetrathyridia of *M. corti* can generate energy under both aerobic and anaerobic conditions [20], which seems to be an adaptation to their ability to invade the host´s niches with various oxygen tensions. In mice, two main predilection sites where larvae multiply asexually are the liver and peritoneal cavity [21]. Whereas in the liver of mice the physiological oxygen concentrations are in the range of 5 to 10% [22], values of oxygen tension in the peritoneal cavity are close to zero [23]. Therefore we aimed to compare the concentration- and time-dependent effects of individual FLs on larval viability, motility, morphological and surface alterations, as well as on the concentrations of the main nutrients glycogen/glucose and neutral lipids after cultivation under hypoxic and aerobic in vitro conditions.

## 2. Results

### 2.1. Assessment of Morphology and Motility

We recorded the motility of larvae in all the tested groups and the data expressed as motility scores are summarized in Table 1. There were no visible differences in the intensity and character of movement in the control groups cultured under the different sets of conditions, and this normal movement was given the score 3+. Under aerobic conditions, SB and SCH moderately decreased motility, the greatest decrease was after treatment with 50 µM of SCH. In contrast, DHSB had a suppressing effect at lower concentrations but over-stimulated the characteristic motility of highly elongated larvae at 50 µM (score 4+). The intensity of shortening/elongation of larval bodies during cultivations with SB and SCH under hypoxic conditions was similar to the movement under aerobic conditions, however DHSB at a higher concentration slowed the intensity of bending and rotating of the bodies of permanently elongated larvae (score 1+ or 0.5+), After 72 h we also monitored the number of dead larvae and the mean proportions of the total counted (%) are summarized in Table 2. The strongest larvicidal effect was with 50 µM of DHSB under hypoxic conditions, which caused a disturbance in the integrity of the tegument followed by the total disruption of larval bodies in 17.5 ± 3.5% of larvae.

### 2.2. Metabolic Activity of Larvae

The MTT test is widely used to study the cytotoxic effect of compounds on cells, and is often used to study the effects of helminth parasites on viability/respiration. Here we tested whether the mitochondrial respiration of larvae cultivated under aerobic and hypoxic conditions can influence the effects of individual FLs on metabolic activity. We evaluated the time-dependent effect of SB, SCH and DHSB at 5 and 50 µM concentrations under aerobic (Figure 2A–C) and hypoxic conditions (Figure 2D–F). The results are expressed as the proportions (in %) of the absorbance corresponding to the amount of extracted formazan in treated larvae versus the untreated control (100%). Under aerobic conditions SB, SCH and DHSB at both concentrations significantly decreased the viability of larvae at each time-point, except for 4 h exposure to 50 µM DHSB. Under hypoxic conditions, after a short incubation time (4 h), 5 µM SB and SCH decreased metabolic activity more effectively than 50 µM, and after a longer exposure of larvae to both FLs (24 h, 72 h), the higher concentration of both FLs exerted a stronger inhibitory effect. We also compared absorbance data for larval samples (normalized for 50 mg) which were cultured with the same concentration of individual FLs and under different cultivation conditions. Cultivation conditions only moderately influenced the effects of individual FLs on the metabolic activity of larvae. 

Fifty µM DHSB exhibited the opposite effect on metabolic activity to SB and SCH at each incubation interval. The increased metabolic activity in larvae cultured under hypoxia with 50 µM DHSB was in contrast with the effects of DHSB during aerobic cultivation. This might be a result of the overstimulation of enzymatic systems of complex II in mitochondria, leading to their collapse and resulting in an uncontrolled reduction of MTT in dead larvae. A high accumulation of formazan in dead larvae was observed under the microscope (not shown).

### 2.3. Neutral Red Uptake

It was shown that viable eukaryotic cells will uptake neutral red dye by active transport, and incorporate it into lysosomes [24]. In this study we adopted this assay to evaluate the effect of SB, SCH and DHSB at concentrations of 5 µM and 50 µM after aerobic and hypoxic cultivation for 72 h. Data for treated larvae are expressed as % of absorbance of untreated control (100%) and are shown in Figure 3A for aerobic and Figure 3B for hypoxic cultivation. The basal rate of neutral red uptake by the larvae was significantly higher under hypoxic than aerobic conditions (not shown). Under aerobic conditions, SB and SCH at low concentrations moderately stimulated the uptake of dye via tegument, having a minimal effect at 50 µM. A significant accumulation of dye was observed in larvae after incubation with 50 µM DHSB (161.55 ± 6.39), indicating alterations in the transport functions of tegument rather than stimulation of the viability of whole larvae.

After the cultivation of larvae under hypoxia, SB at a low concentration increased the uptake of dye (151.63 ± 19.90 in comparison with control, *p* < 0.01), but had no significant effect at 50 µM. SCH did not modulate the transport of dye significantly. After treatment with 50 µM DHSB, the concentration of neutral red in larvae was significantly elevated (139.58 ± 12.29%), but to a lower extent than was found in larvae after aerobic cultivation (*p* < 0.05). Representative images of larvae from hypoxic cultivations exposed to FLs and the distribution of neutral red are shown in panel C of Figure 3. In control larvae, dye was accumulated at a higher rate by the posterior part than by the anterior part of larvae. Larvae from the neutral red assay conducted under aerobic conditions had a similar distribution of dye and are not shown.

### 2.4. Concentration of Glucose in Larvae and Glycogen Localization

Glucose is the major nutritional source for helminths and is actively transported by cestodes via the tegument [25]. Therefore, we evaluated whether FLs can influence the concentration of glucose deposited in larvae in the form of glycogen. We showed that all three FLs significantly modulated the glucose concentration in *M. vogae* larvae under aerobic (Figure 4A) and hypoxic conditions (Figure 4B). Freshly isolated larvae contained 47.01 ± 2.44 mg glucose in 1 g of wet weight tissue. The transfer of larvae from mice to in vitro cultivation systems resulted in a significant increase in glucose concentrations to 89.23 ± 14.40 mg/g under aerobic conditions and 72.24 ± 2.74 mg/g under hypoxic conditions. At the lower concentration, SB elevated glucose concentration, however at 50 µM, a significant lowering was found with both cultivation systems compared to the control. Treatment with SCH resulted in a reduction in glucose in both cultivation systems, more at the higher concentration (66.26 ± 6.37 vs. 43.65 ± 1.46). The most dramatic effect was seen with DHSB, which caused a reduction in glucose at concentrations of 5 µM (39.83 ± 6.92) and 50 µM (11.86 ± 1.22) under hypoxic conditions.

Glycogen was localized on cryostat sections of larvae as dark-purple granules by PAS staining, and the staining pattern was very similar for larvae from both cultivations. Figure 4C shows representative images of glycogen localization after exposure to 50 µM of FLs under hypoxic conditions only. In larvae from control cultivation and much less in SB and SCH-treated larvae, glycogen exhibited discrete localization in the sub-tegumental parts of larval bodies and in the suckers. In larvae treated with DHSB, glycogen depots were not detected in the sections. The pink stains seen in the central part of larval bodies were possibly other sugar-containing molecules.

### 2.5. Neutral Lipid Concentrations and Their Distribution

Neutral lipids, comprised mainly of triacylglycerol, cholesterol and free fatty acids, can also serve as energy stores for helminths [26]. Here we monitored the effects of FLs (5 µM and 50 µM concentrations) under both cultivation conditions on the concentration of total neutral lipids after 72 h. Freshly isolated larvae contained 55.19 ± 3.2 mg lipids/g of wet weight larval tissue and, in contrast with glucose, their content dropped significantly with the aerobic cultivation of larvae (38.13 ± 4.27) (Figure 5A) and more under hypoxic conditions (28.28 ± 5.2) (Figure 5B). The low concentration of all three FLs only moderately affected the content of neutral lipids in larvae. The presence of 50 µM of SB and SCH in the medium resulted in a significant reduction in lipids in larvae cultivated under both aerobic and hypoxic conditions compared to the untreated control. Interestingly, DHSB exhibited the opposite effects on this parameter with respect to oxygen availability and thus metabolic pathways. It significantly reduced lipids (*p* < 0.001) under aerobic conditions (16.59 ± 4.19), whereas it elevated its concentration after hypoxic cultivation (49.57 ± 10.21), correlated with a markedly reduced larval motility.

The localization of lipid droplets and nuclei on whole-mount preparations of larvae was examined by confocal scanning laser microscopy after histochemical staining. Very similar staining patterns were observed in control larvae and larvae treated with SB and SCH (50 μM) under aerobic and hypoxic conditions, which is shown in the selected images (upper panel). Representative images (lower panel) of larvae after DHSB treatment (50 μM) exhibited different staining patterns for individual cultivations. In contrast to glycogen, lipid droplets were localized mainly in the central parenchymal part of larval bodies and below the suckers in larvae from the control cultivation. Changes in the concentrations of lipids after exposure to FLs correlated with the distribution and fluorescence intensity of lipid droplets, being the most affected in the neck regions.

### 2.6. Scanning Electron Microscopy of Larvae

The shape of the body and effect on the tegument of larvae after three days of incubation with 50 µM of FLs were analyzed by scanning electron microscopy, and representative images are shown in Figure 6. The control larvae (CTRL-1, 2) had a typical elongated shape and the body merged smoothly into the scolex with four suckers. Longer microvilli localized to the anterior part of the tegument were intact, as were short microvilli localized to the posterior part of larval bodies. Larvae in SB- and SCH-treated groups seemed to have a normal shape, and the tegumental microvillous surface appeared to be normal on both parts. Larvae exposed to DHSB under both cultivation conditions had highly elongated bodies, an extended narrow neck region and protruded scolex, which was accompanied by the formation of surface protuberances in some parts. Erosions of the microvillous layer of the tegument were prominent in the posterior part of larvae and much less in the anterior.

## 3. Discussion

Larvae of *Mesocestoides vogae* are characterized by asexual growth in rodents and lizards [27], and proliferative stages of this species and also of other *Mesocestoides* spp. were found to be responsible for serious infections in dogs and cats with poor prognosis [28,29,30]. They can be easily maintained in vitro under aerobic or anaerobic conditions, and are metabolically active even in serum-free and axenic cultivation systems [31]. From this perspective, they appear to be a versatile model for pharmacological studies.

Helminths have exploited a variety of energy transducing systems in their adaptations to the specific habitats in their hosts, and the most important factors are the nutrient and oxygen supply [32]. Several studies showed that certain developmental stages of helminths residing in an anaerobic environment, mostly adults, possess a unique anaerobic respiratory system called the NADH-fumarate reductase system, which couples mitochondrial complex I and II, generating ATP even in the absence of oxygen (see for review [33]). Kӧhler and Hanselmann [20] showed that *M. vogae* larvae are metabolically active in both an aerobic and anaerobic/hypoxic microenvironment. This and our findings indicate that they can utilize the NADH-fumarate reductase system and mitochondrial respiration involving NADH-oxidase localized in complex I, an enzyme involved in aerobic respiration described in mammals. Therefore, in this in vitro study we evaluated the activities of SB, SCH and DHSB under aerobic cultivation conditions and in a cultivation system with a reduced external supply of oxygen/CO_2_, partially resembling hypoxic environments in hosts. Firstly, we found that larvae transferred from mice to in vitro cultivation systems adapted to the new microenvironments by gradually decreasing metabolic activity, neutral red uptake and concentrations of neutral lipids in contrast with significantly elevated glucose content, without visibly affecting their viability and motility. These changes in larval physiology due to in vitro cultivation conditions, which differ from those in their natural habitats in the hosts, have to be considered in the evaluation of the effects of the compounds.

In this study we found that SB and SCH did not modify motility at low concentrations (5 μM) and moderately reduced the frequency of body contractions at high concentration (50 μM) (Table 1). The effects were similar over the 72 h period of cultivations for the aerobic and hypoxic culture systems, and no irreversible morphological alterations or changes in body shape were observed. In contrast, exposure to DHSB, more profoundly at the high concentration, caused an elongation and flattening of larval bodies within a few hour after exposure, and larvae lost the ability to contract their whole bodies in the anterior-posterior direction. Under aerobic conditions, DHSB treatment stimulated the movement, characterized as the frequent bending and rotating of the whole body. Under hypoxic conditions, this compound highly reduced movement, correlating with the highest percentage of dead larvae compared to the control culture. The physiological motility of *M. vogae* larvae is a process coordinated by multiple neurotransmitters and neuropeptides and other molecules (for example Hrčkova et al. [34]), which act on the body muscle system formed of longitudinal and transversal smooth muscles [35]. They showed that muscles are attached to the subtegumental region, thus influencing the body shape. The tegument of flatworms is a very flexible dynamic structure responsible for the absorption and active transport of nutrients and other molecules from the hosts [36,37]. We suppose that DHSB not only caused irreversible changes in the molecular structure of the tegument, but also profoundly disturbed the physiological homeostasis in larval cells, leading eventually to their death. The opposite effects of DHSB on motility in terms of oxygen availability in cultures indicate that these effects could be linked to aerobic and anaerobic energy metabolism and respiration.

We further tested the activities of FLs on these processes by means of the MTT test, in which the reduction of MTT to formazan during mitochondrial respiration is catalyzed by succinate dehydrogenase using succinate as an electron donor or by the activity of NADH dehydrogenase [36,37]. Kӧhler and Hanselmann [20] demonstrated that larvae of *M. corti* cultured under anaerobic conditions produce more succinate, suggesting the presence and high activity of an NADH-fumarate reductase system. In this study, with aerobic cultivation, the metabolic activity of larvae was suppressed by the low concentration of each of the FLs, already within 4 h of exposure, but the most after 72 h of cultivation with the 50 μM concentration. In accordance with the previous results, the effects of DHSB under hypoxic conditions differed from that of SB and SCH, and a high amount of formazan was extracted from larvae after exposure to the 50 μM concentration. Using the enriched mitochondrial fraction, Matsumoto et al. [38] demonstrated that the NADH-fumarate reductase system is the predominant respiratory chain in protoscoleces of the closely related species *E. multilocullaris*, indicating that parasite mitochondria are highly adapted to anaerobic environments. So far, no such study on the enzyme activities that form mitochondrial respiratory systems in *M. vogae* larvae has been conducted. The decreased metabolic activity caused by SB and SCH, regardless of the cultivation conditions, could imply that these FLs interfere with the functions of the enzymes, probably dehydrogenases, involved in both aerobic and anaerobic respiration. The significant increase in reduced formazan after DHSB treatment over the control might be due to the overstimulation of dehydrogenases or uncontrolled release of reducing substrates in damaged/dying larvae. The different effects of individual FLs on larval motility, physiology and respiration could be linked to their chemical properties, which determine the interactions with proteins, nucleic acids and lipids. As in other polyphenols, all three FLs are lipophilic in the order DHSB > SB> SCH [39], which enables their rapid penetration into lipid membranes. SB and DHSB were shown to display high in vitro affinities for direct binding to P-glycoprotein in cell membranes, which are involved in accelerated drug efflux from the cells, making them resistant to drugs [40]. Using molecular modelling methods, Kubala et al. [41] identified SCH and its dehydro derivatives as potent inhibitors of sodium-potassium adenosine triphosphatase (the sodium pump) in membranes, an enzyme of crucial importance for all animal cells. Moreover, DHSB was shown to exert a stronger direct antioxidant and also anticancer effect than SB [42].

The internalization of neutral red dye in eukaryotic cells is an active process mediated via pinocytosis [43] and/or phagocytosis in myelo-monocytic cells [24]. By measuring the rate of dye internalization via the larval tegument, we demonstrated that the low concentration of SB, SCH and DHSB enhanced this process in larvae in both cultivation systems. The high concentration of SB and SCH had only a minimal effect, however the rapid accumulation of dye in larvae after exposure to 50 μM of DHSB within 1 h could probably be linked to the profound alterations in the protein transporters in the tegument. Similarly, another lipophilic flavonoid, genistein (0.5 mg/mL), significantly disturbed the activities of several tegumental enzymes in the cestodes *Raillientina echinobothrida* [6] and *Echinocossus* spp. [8]. By using scanning electron microscopy, we showed that a dense layer of longer microtriches localized to the anterior end of larvae was more resistant to the treatments with FLs than the shorter microtriches at the posterior end. Here the most intense damage was seen after exposure to 50 μM of DHSB, in the form of aneurysms of the tegument and loss of the microvilli-like surface. Similar elongation and flattening of larval bodies, blebs, holes and loss of microtriches after in vitro exposure to high doses (150–250 µg/mL) of the essential oil thymol were reported by Maggiore and Elissondo [44].

Glycogen is a carbohydrate complex formed predominantly of glucose units, and serves as the most important energy reserve in both the larval and adult stages of cestodes [32]. After PAS staining, glycogen was seen in the sections of control larvae as dark purple granules in subtegumental parts. No glycogen granules were observed in larvae after exposure to 50 μM of DHSB. Of the natural compounds with antiparasitic effects examined, arthemeter and genistein were also found to decrease glycogen concentration in flatworms [45,46]. It is established that all eukaryotic cells, including flatworms, import glucose across their hydrophobic surface using enzymatic glucose transporters, mostly ATPases [47]. Zhan et al. [48] demonstrated on several model cell lines that SB and DHSB dose-dependently reduced basal and insulin-dependent glucose uptake by inhibiting glucose transporters. Interestingly, all three FLs significantly reduced the concentration of glucose in larvae, of which DHSB was the most effective under hypoxic conditions. These findings might indicate that FLs reduced the uptake of glucose from the RPMI medium by inhibiting glucose transporters, however, effects on glucose catabolism could also be considered. Reduced glucose in larvae after SB and SCH treatment and significantly reduced activities of enzymes forming respiratory chains in mitochondria imply an imbalance in energy metabolism, though insufficient to cause larval death. With DHSB the dramatic reduction in glucose, as well as the excess formazan production in larvae cultured under hypoxic conditions, could be result of the disruption of enzymatic systems involved in generating energy in the form of ATP. With eukaryotic cells, Detaille, et al. [49] found that silibinin dose-dependently reduced glycolysis from carbohydrates in perfused rat hepatocytes via an inhibitory effect that targeted pyruvate kinase activity. Furthermore, a dramatic effect upon oxidative phosphorylation was shown, as evidenced by a fall in ATP-to-ADP ratio, together with an increase in lactate to-pyruvate ratio.

Results from various experiments suggest that helminths are unable to synthesize sterols (cholesterol) and fatty acids de novo [32] and depend largely on the utilization of host lipids during infection to survive. As for composition, for example in the cestode *Taenia taeniaeformis*, the isolated lipid droplets mainly consisted of neutral lipids with triglycerides, sterol esters, sterols and free fatty acids being the major components [50]. In this study, the concentration and localization of neutral lipids in larvae under both cultivation conditions were also determined. In cestoda, the binding and transport of neutral lipids is ensured by ligand-binding proteins that are confined to both the cytoplasm and tegument [51]. Information about lipid catabolism in the larval stages of cestodes is limited, but in other helminths it is regulated in a similar way to the system in their hosts [26]. We found a decline in lipids stores in control larvae in comparison with basal levels, stronger under hypoxic conditions. At the low concentration, all FLs only had a minor effect on this type of nutrient, but the high concentration of SB and SCH significantly reduced lipids stores regardless of oxygen availability in cultures. In contrast, DHSB decreased lipids under aerobic conditions but significantly elevated them under hypoxic conditions. It was reported that in helminths, glycerol can eventually enter glycolytic pathways for further degradation [26], and enzymes of β-oxidation were found in the trematode *Schistosoma mansoni*, which was confirmed by Huang et al. [52], though fatty acids are normally catabolized via the TCA cycle. We suppose that the observed changes in lipids stores in larvae could be due to interference with lipids transporters in the tegument as well as the effect on lipid metabolism.

## 4. Materials and Methods

### 4.1. Silymarin Flavonolignans and Other Chemicals

Silymarin was purchased from Liaoning Senrong Pharmaceutical Co. (Panjin, China, batch no. 120501). Silybin (a natural equimolar mixture of diatereomers A and B) and silychristin were isolated from silymarin as described previously [53]. Briefly, a portion of silymarin was suspended in methanol and after brief stirring, undissolved silybin was filtered off. The filtrate was concentrated and loaded into a column with Sephadex LH-20 (Sigma Aldrich, Darmstadt, Germany) and eluted with methanol to obtain silychristin A. Silybin was used as a natural equimolar mixture of two diastereomers with purity of over 95% (HPLC PDA), silychristin A contained about 3% (HPLC PDA) silychristin B and its purity was over 95% (HPLC PDA). Racemic 2,3-dehydrosilybin was prepared from silybin in the same way as described previously [40] by oxidation with iodine in acetic acid and its purity was over 95%. 

The producers of chemicals and kits are indicated in the text, except for those purchased from Sigma Aldrich, Darmstadt, Germany): MTT (3-(4,5-dimethylthiazol-2-yl)-2,5-diphenyltetrazolium bromide), Neutral Red powder, d-glucose, Nile Red powder, glyceryl trioleate (Triolein), DABCO (1,4 diazabicyclo(2,2,2)octane, amphotericin B, paraformaldehyde (PFA). RPMI-1640 with stable glutamine and HEPES, PenStrep (penicillin/streptomycin), bovine foetal serum, Hoechst 33433, anthrone, glutaraldehyde (10% solution) and DMSO were purchased from Merck (Darmstadt, Germany).

### 4.2. Parasites and In Vitro Culture System

Infection with *M. vogae* tetrathyridia was maintained in an outbred ICR strain of mice by the intraperitoneal passage of larvae isolated from chronically infected mice (3–5 months of age). Maintaining the infection in ICR mice was approved by the ethics committee of the State Veterinary Administration of the Slovak Republic with reference number Ro-3871/15-221c. For each in vitro experiment, larvae were aseptically isolated from the peritoneal cavity of an infected mouse, washed several times with sterile PBS and larval inoculation samples were prepared in a volume of 0.4 mL for each cultivation, instead of counting the larvae. This volume of settled larvae contained approximately 3.0–3.5 × 10^3^ larvae. The larvae were then transferred into 50 mL cell culture flasks (Becton Dickinson, Franklin Lakes, NJ, USA) with a plug seal cap for hypoxic conditions or with a ventilated cap for the cultivations in aerobic conditions (5% CO_2_ and access to atmospheric oxygen) at 37 °C. Individual experimental flasks were then filled with 10 mL of RPMI 1640 medium supplemented with 5% bovine foetal serum, 1% PenStrep antibiotic solution, 0.1% gentamycin (Lonza, Basel, Switzerland) and 0.1% amphotericin B. Stock solutions of silybin (SB), silychristin (SCH) and dehydrosilybin (DHSB) were prepared in 100% DMSO. At the concentration of 50 μM, the culture medium contained 0.1% DMSO, which had no effect on larval viability and motility. One control flask (medium with corresponding DMSO concentration) and cultures of larvae treated with SB, SCH and DHSB, respectively, were set in each in vitro experiment. Medium in flasks was replaced with fresh medium containing the corresponding concentration of individual FLs once after the first 36 h.

### 4.3. Quantification of Effects of FLs in Larval Cultures

A simple procedure for the quantitative assessment of the effects on a given amount of larvae in all assays was established. As larvae are of various sizes and multiplication stages, throughout this study wet weight of larval mass was determined instead of counting them, and was done in pre-weighed 1.5 mL test tubes. Briefly, at the selected time of incubation, larvae were aseptically transferred from the culture flask to pre-weighed tubes in a volume of approximately 50 μL and left to settle. After removing the liquid, tubes were weighed and the wet weight of larval sample was calculated, being in the range of 20 to 35 mg. For each assay, individual parameters were first calculated for the precise weight of larvae in each sample tube and then re-calculated for the selected amount of wet larval tissue (50 mg or 1 g). In each biochemical assay and under each experimental set of conditions, triplicate larval samples were used for each treatment.

### 4.4. Metabolic Activity of Larvae

Metabolic activity of larvae was evaluated by the MTT test, which we initially used to assess the time-dependent effects of SB, SCH and DHSB at concentrations of 5 μM and 50 μM after 4 h, 24 h and 72 h of cultivation. Larvae exposed to FLs were transferred to pre-weighed tubes for hypoxic conditions and to 24-well plates for aerobic cultivation in 1 mL of RPMI medium. 4 h before the selected period of exposure, 40 μL of 0.5 % MTT was added to the tubes/plates. The supernatant was then removed, larvae were washed with PBS and homogenized with a pestle and mortar in 200 μL of DMSO. After centrifugation at 3000 rpm for 10 min, DMSO containing formazan was transferred to a 96-well plate and absorbance was measured at 550 nm with the reference filter at 630 nm using a Multiscan FC Plate Reader (Thermo Scientific, Waltham, MA, USA). The absorbance obtained for the real weight of larval tissue in tubes was re-calculated for 50 mg of tissue/sample, and these values were used to express changes in metabolic activity relative to untreated control samples (100%) for each time point and concentration.

### 4.5. Neutral Red Uptake

Neutral red uptake is a common quantitative assay to determine cell viability and toxic effects of the compounds in vitro. This method was used to examine larval viability in freshly isolated larvae and in larvae after 72 h of incubation with 5 μM and 50 μM of SB, SCH, DHSB as well as in untreated controls under both types of conditions. In the assay we used the In vitro Toxicology Assay Kit Neutral Red Based (Sigma-Aldrich, Darmstadt, Germany) with some modifications. Briefly, larvae were transferred from flasks into 1.5 mL tubes (in triplicates/group) and filled with 1.0 mL of fresh RPMI medium containing 20 μg/mL of neutral red. Incubation involved mild shaking at 37 °C and was terminated after 60 min. After washing the larvae, they were homogenized in 1.5 mL of solubilization solution. The supernatant containing neutral red was collected into another tube after centrifugation. The optical density (OD) of the samples was measured at 540 nm in a Specord S600 spectrophotometer (Analytic Jena, Jena, Germany). Concentrations of neutral red expressed as OD for the actual amount of larval tissue were re-calculated for 50 mg of tissue in each sample, and these values were used to calculate the viability of larval cells (in %) relative to untreated control samples (100%) at each time point.

### 4.6. Motility of Larvae and Morphological Alterations of Larvae

The larval stage of *Mesocestoides* spp. (tetrathyridium) is an organism with a size between 0.1 to 2 mm, which is morphologically formed of unsegmented parenchymal tissue, scolex and well developed muscle and nervous system [34,35]. Under physiological conditions at 37 °C, the movement of larvae appears as a gradual shortening followed by elongation of larval bodies resembling contractions in the anterior-posterior direction due to its flexible tegument and muscle system. We monitored the character and intensity of movement (number of contractions/min) of at least 100 individual larvae/flask directly in the culture flasks using an inverted microscope (Leica, Wetzlar, Germany) and expressed it as a motility score. Larvae under control cultivations were assigned a score of 3+, having typically 5–10 contractions/min of whole bodies per min. The movement of larvae exposed to FLs was assessed as follows: 0.5+, minimal movement in the majority of irreversibly elongated larvae, 1+, low frequency (2–4/min) of shortening/elongation of normal larval bodies in only a part of the larval tissue, 2+, more frequent contractions of whole larval bodies (3–7/min) in the majority of larvae, 4+, intense movement appearing as bending and rotating of the bodies of permanently elongated larvae, which were only capable of minimal shortening/elongation in the anterior-posterior direction. The final motility scores represent data from two different experiments.

### 4.7. Glucose Concentration and Localization of Glycogen in Larvae

The concentration of free glucose and glycogen-derived glucose in the larval tissue was quantified by the anthrone method with some modifications. Anthrone reagent was prepared freshly each time by dissolving 20 mg of anthrone in 10 mL of 96% H_2_SO_4_. The concentration of glucose was measured in larvae isolated from mice (basal) and in larval samples exposed to FLs (5 μM and 50 μM of each FLs) for 72 h. Larval samples, with pre-determined weight as indicated above, were transferred into glass tubes and hydrolyzed in 4 mL of 10% KOH in a boiling water bath for 15–20 min. 0.5 mL of the hydrolysate was slowly mixed with 2 mL of anthrone reagent in an ice bath. The reaction mixture was boiled for exactly 10 min, cooled rapidly and used for the spectrophotometric determination of glucose at 630 nm. Simultaneously, a standard curve using d-glucose as standard in 10% KOH was prepared over the range of 0.2 to 1.0 mg/mL. Glucose in experimental samples was then calculated for 1 g of larval tissue/sample.

The localization of glycogen and molecules containing saccharides, for example glycoproteins, was examined in cryostat sections of larvae after treatment with 50 μM of FLs. Larvae were immediately embedded into NEG 50^TM^ (Thermo Scientific) fluid for frozen tissue and 12 μm-thick cryosections were cut in a Cryocut 1800 (Leica). Sections were then mounted on slides coated with 2% gelatin solution, fixed with 4% PFA for 10 min, washed in PBS, stained with periodic acid-Schiff (PAS) staining system (Sigma-Aldrich, Darmstadt, Germany) and observed under a light microscope (Olympus, Prague, Czech Republic).

### 4.8. Neutral Lipids Concentration

Nile red is a fluorescent stain for the detection of cytoplasmic lipid droplets in eukaryotic cells of animal as well as plant origin [54]. We developed the assay for the quantification of neutral lipids in larval tissue. Freshly isolated larvae and larvae after 72 h of cultivation under both sets of conditions were weighed and homogenized mechanically in 0.2 mL of 50% DMSO for 10 min. The fluorescent determination of lipid content was performed in 96-well black plates (Greiner-Bio, Kremsmünster, Austria) and the best assay-carrier solvent, giving the minimal background fluorescence, was found to be 5% DMSO in water. The assay was run in 100 µL of reaction mixture containing 90 µL assay solvent and aliquots (10 µL) of dispersed homogenates of larvae. Based on our preliminary testing, the determination of neutral lipids in the homogenates using nile red was more accurate than in the supernatants due to the limited solubility of lipids in 5% DMSO. The lipid standard for preparing the calibration curve was Triolein (TO) and its stock solution (1 mg/mL) was made in isopropyl alcohol. The same carrier solvent was used to make seven dilutions (1:1) of TO from 100 µg/0.1 mL/well to 1.56 µg. A standard curve (in carrier solvent) was prepared before adding 2 µL/well of nile red dye diluted in DMSO to a concentration of 100 µg/mL. The same volume of nile red was added to sample wells and, after vigorous shaking of the plate, fluorescence was read at 530 nm (excitation) and 604 nm (emission), providing the maximum fluorescence. The concentration of lipids was determined for the real wet weight of larval tissue in tubes and then calculated for 1 g of tissue.

### 4.9. Confocal Laser Scanning Microscopy for Localization of Neutral Lipids

The affinity of nile red to neutral lipids was employed to study the localization of lipid droplets in larvae after 72 h of cultivation with 50 µM FLs. Specimens were fixed in 4% PFA in PBS (pH 7.2) for 4 h and processed for whole-mount histochemistry. After washing in PBS, larvae were immersed in the staining mixture containing 10 µg/mL nile red in PBS (stock solution was 1 mg/mL DMSO) and also the fluorescent dye Hoechst 33433 at a concentration of 5 µg/mL of PBS, which allowed the visualization of nuclei in blue. Larvae were stained for 5 h at 8 °C in the dark and after washing, they were mounted on slides in glycerol solution containing 2.5% DABCO. The distribution of neutral lipid droplets was visualized using a LSCM Olympus Fv1000 confocal scanning laser microscope (Olympus, Prague, Czech Republic) and analyzed with a FV10-ASW 2.0 Viewer (Olympus, Prague, Czech Republic).

### 4.10. Scanning Electron Microscopy of Larvae

Tegumental changes in the tetrathyridia following in vitro incubation with 50 µM were examined by scanning electron microscopy (SEM). Specimens were washed in warm PBS and immediately fixed with 2.5% glutaraldehyde in PBS buffer overnight at 4 °C and stored at the same temperature before further processing. Samples were washed three times in PBS and dehydrated through an alcohol series (25%, 50%, 75%, 90% and 96%). The samples were then transferred into porous specimen pots (Quorum Technologies Ltd, Ringmer, UK), treated twice in 100% ethanol for 20 min and critical point dried with liquid CO_2_ in a K 850 unit (Quorum Technologies Ltd., Laughton, UK). The dried samples were mounted onto Die-Cut Carbon Conductive Double-Sided Adhesive Discs (Structure Probe, Inc., West Chester, PA, USA) and sputter-coated with 20 nm of gold in a Polaron Sputter-Coater (E5100) (Quorum Technologies Ltd.). The final samples were examined in a FEI Nova NanoSem 450 scanning electron microscope (FEI, Brno, Czech Republic) at 5 kV using secondary electron detectors (SE or TLD) or the back-scattered electron detector (CBS).

### 4.11. Statistical Analysis

Statistical analysis of data was performed using GraphPad Prism version 7.00 for Windows (GraphPad Prism^®^ Software, Inc., San Diego, CA, USA). Results were expressed as mean values ± standard deviation (SD) and were calculated from triplicate samples/group obtained from three independent in vitro experiments. For analysis, one way ANOVA, followed by Tukey’s post-hoc test was used and statistical differences were evaluated between mean values in the control and treated groups and in indicated assays also between basal and control groups. *p* < 0.05 were considered statistically significant.

## 5. Conclusions

In summary, we demonstrated that SB and SCH significantly suppressed the functions of mitochondrial enzymes, reduced glycogen/glucose and neutral lipid stores, thereby inducing a physiological misbalance, which however did not lead to the death of larvae. In contrast, DHSB exhibited a concentration- and time-dependent larvicidal effect due to marked damage to the tegument and probably protein transporters, as well as the complete disruption of larval respiration, physiology and metabolism.

## Figures and Tables

**Figure 1 molecules-23-02999-f001:**
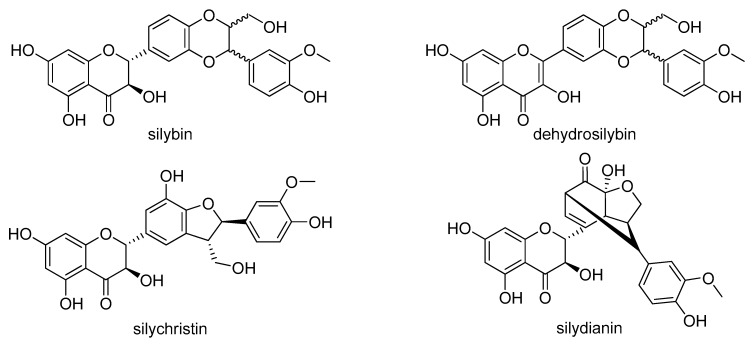
Selected silymarin flavonolignans.

**Figure 2 molecules-23-02999-f002:**
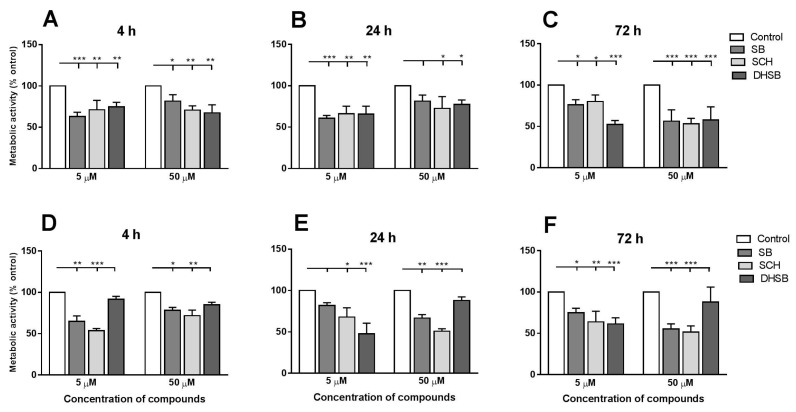
Concentration- and time-dependent effect of SB, SCH and DHSB on metabolic activity of larvae under aerobic (**A**–**C**) and hypoxic (**D**–**F**) conditions. Larvae were treated with 5 μM and 50 μM FLs, and MTT was added 4 h prior to formazan extraction. The results represent means ± SD from three experiments done in triplicates/group and are expressed as the proportions (%) of absorbance in treated groups related to the absorbance in the control group (100%). Significantly different values from controls are shown as * *p* < 0.05, ** *p* < 0.01, *** *p* < 0.001. (SB) silybin; (SCH) silychristin; (DHSB) dehydrosilybin.

**Figure 3 molecules-23-02999-f003:**
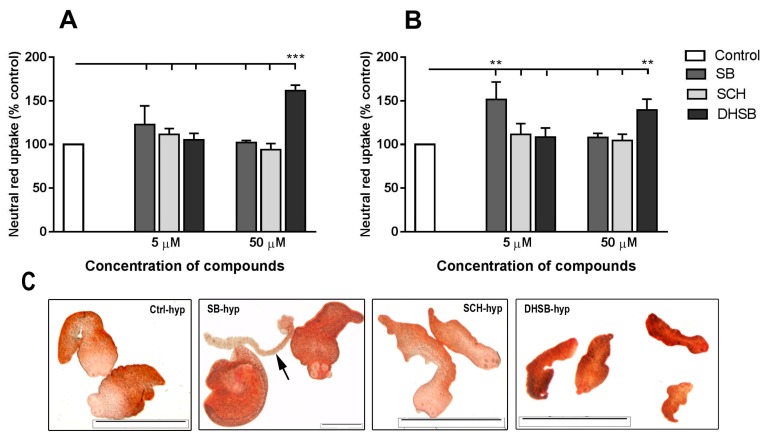
Viability of larvae measured by means of neutral red uptake after 72 h of incubation under aerobic (**A**) and hypoxic (**B**) conditions with SB, SCH and DHSB (5 µM and 50 µM). Results represent means ± SD from three experiments in triplicates/group and are expressed as the proportions (%) of absorbance in treated groups relative to the absorbance in the control group (100%). Significantly different values from controls are shown as ** *p* < 0.01, *** *p* < 0.001. (**C**) Representative images of live larvae from hypoxic cultivation after exposure to 50 µM concentration of FLs. Note dead larva (arrow). Scale bar = 2000 µM, Ctrl (control), (SB) silybin; (SCH) silychristin; (DHSB) dehydrosilybin.

**Figure 4 molecules-23-02999-f004:**
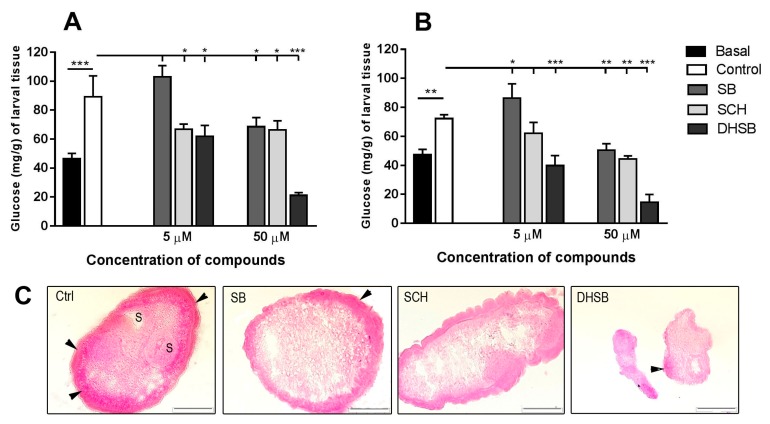
Concentration of glucose in larvae after isolation from mouse (basal), in control group and after treatment with SB, SCH and DHSB at 5 μM and 50 μM concentration for 72 h under aerobic (**A**) and hypoxic (**B**) conditions. Significantly different values from corresponding controls are shown as * *p* < 0.05, ** *p* < 0.01, *** *p* < 0.001. (**C**) Representative images of cryosections of larvae from hypoxic incubation after exposure to 50 μM concentration of FLs. Glycogen (arrows) was mostly localized as dark purple granules in sub-tegumental parts of control, SB and SCH-treated larvae, and was lacking in DHSB-treated larvae. Scale bar = 100 μm, Ctrl (control), (SB) silybin; (SCH) silychristin; (DHSB) dehydrosilybin.

**Figure 5 molecules-23-02999-f005:**
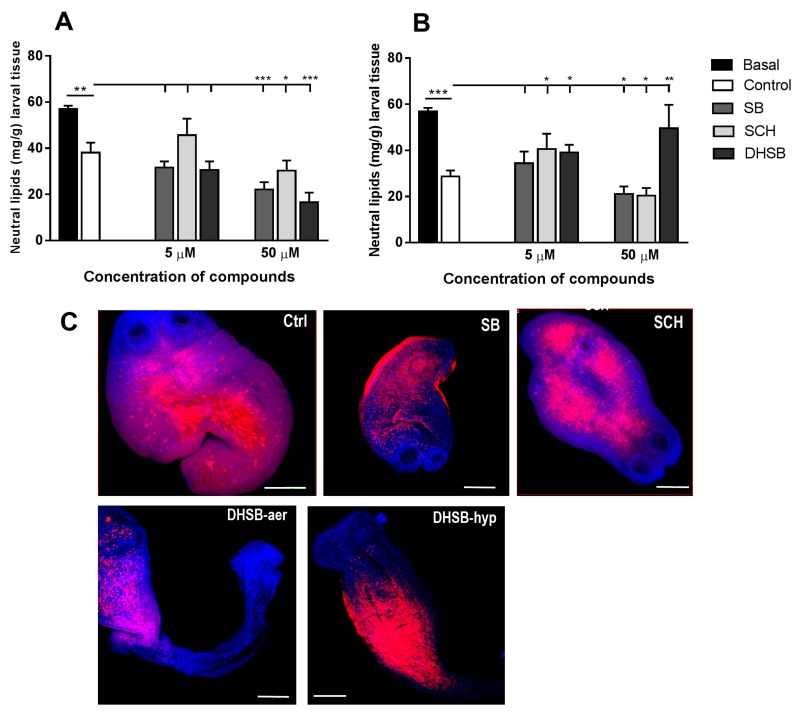
Neutral lipids concentration in larvae after isolation from mouse (basal), in control group and after in vitro treatment with SB, SCH and DHSB at 5 μM and 50 μM concentrations for 72 h under aerobic (**A**) and hypoxic (**B**) conditions. Whole-mount histochemistry and confocal laser scanning microscopy were used to demonstrate the localization of neutral lipids (red) and nuclei (blue) in larvae (**C**). Images show distribution pattern of lipids in control larvae and after treatment with 50 μM of SB and SCH (upper panel), identical for both aerobic and hypoxic cultivations. Representative images of larvae after DHSB treatment (50 μM, lower panel) showed different staining patterns for individual cultivations. Scale bar = 100 μm, (SB) silybin; (SCH) silychristin; (DHSB) dehydrosilybin. Significantly different values from corresponding controls are shown as * *p* < 0.05, ** *p* < 0.01, *** *p* < 0.001.

**Figure 6 molecules-23-02999-f006:**
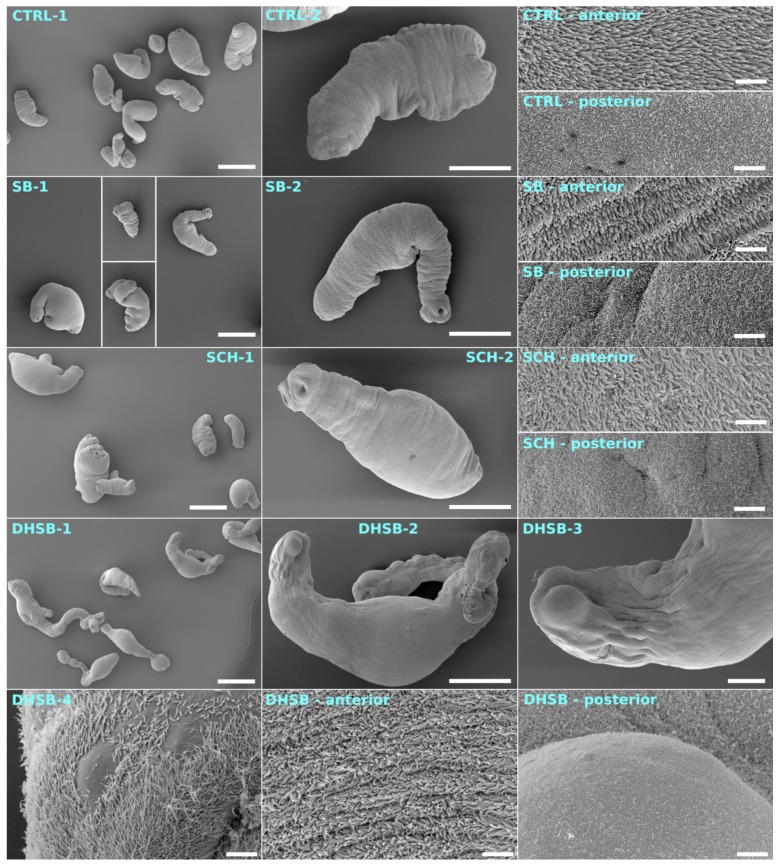
Effect of 50 μM of SB, SCH and DHSB on whole larval morphology and tegumental alterations after 6 days of incubation were analyzed by scanning electron microscopy. Images placed in left panel (1) and middle panel (2) show shape and morphology of larvae in control (CTRL) and treated groups. Images localized in right and bottom panels represent high resolution microphotographs of tegumental surface on anterior and posterior parts of larvae. Elongation, flattening of larval bodies and narrowing of neck as well as severe damage to tegument seen as aneurysms of tegument and loss of microtriches at posterior part were recorded after DHSB exposure. Scale bars are as follows: CTRL-1, SB-1, SCH-1, DHSB-1—500 μm; CTRL-2, SB-2, SCH-2, DHSB-2—200 μm; DHSB-3—50 μm; DHSB-4—5 μm; all posterior and anterior—5 μm.

**Table 1 molecules-23-02999-t001:** Effect of flavonolignans on motility of larvae expressed as motility score. The intensity and mode of larval movement was evaluated over three days of cultivation under aerobic and hypoxic conditions under an inverted microscope. Normal motility is seen at 37 °C as moderate contractions of larval bodies in the anterior-posterior or dorsal-ventral directions.

Tested Compounds	Concentration (μM)	Aerobic Cultivation			Hypoxic Cultivation		
		Motility Score/Days of Cultivation					
		Day 1	Day 2	Day 3	Day 1	Day 2	Day 3
Control	-	3+	3+	3+	3+	3+	3+
SB	5	2+	2+	2+	3+	3+	3+
	50	2+	2+	2+	2+	2+	2+
SCH	5	2+	2+	2+	2+	2+	2+
	50	1+/2+	1+/2+	1+/2+	1+/2+	1+/2+	1+/2+
DHSB	5	1+	1+	1+	1+/2+	1+	1+
	50	4+	4+	4+	1+	1+	0.5+

3+ motility of untreated larvae having typically 5–10 contractions of the whole body per min. 0.5+ minimal movement in the majority of extremely elongated larvae, 1+ low frequency (2–4/min) of shortening/elongation of normal larval bodies in only part of larval tissue, 2+ more frequent contractions of whole larval bodies (3–7/min) in majority of larvae, 4+ intense movement in the form of bending and rotating of the bodies of permanently elongated larvae, which were only capable of minimal shortening/elongation in the anterior-posterior direction. (SB) silybin; (SCH) silychristin; (DHSB) dehydrosilybin.

**Table 2 molecules-23-02999-t002:** Proportions of dead larvae out of total counted larvae (in %) evaluated under inverted microscope after three days of cultivation under both aerobic and hypoxic conditions. Dead larvae were assessed according to the loss of movement and disintegrated tegument.

Tested Compounds	Concentration [μM]	Proportions [%] under Cultivation Conditions	
		Aerobic	Hypoxic
Control	-	1.9 ± 0.8	1.4 ± 0.9
SB	5	1.1 ± 0.7	1.8 ± 0.8
	50	2.4 ± 1.5	3.0 ± 1.2
SCH	5	2.9 ± 2.0	2.9 ± 1.5
	50	5.1 ± 2.3	4.7 ± 2.9
DHSB	5	3.7 ± 2.1	5.4 ± 3.6
	50	7.9 ± 3.3	15.1 ± 4.7

Data are expressed as the proportion of dead larvae out of total counted larvae (%) from two independent experiments. (SB) silybin; (SCH) silychristin; (DHSB) dehydrosilybin.

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
