# Peer review of "Differential Effects of the Flavonolignans Silybin, Silychristin and 2,3-Dehydrosilybin on *Mesocestoides vogae* Larvae (Cestoda) under Hypoxic and Aerobic In Vitro Conditions"

_molecules, 2018, doi:10.3390/molecules23112999_

Reviewer 1 Report

The authors of "Differential Effects of Flavonolignans Silybin, Silychristin and 2,3-Dehydrosilybin on Mesocestoides vogae Larvae (Cestoda) under Hypoxic and Aerobic in vitro Conditions" report on a very complete set of experiments providing a clear and sound view of the effects of 3 different flavonolignans on cestoda larvae under both aerobic and hypoxic conditions.

I strongly recommend this work for publication following (very) minor revisions.

Line55 / Figure 1: silydianin is not shown. Change the text or draw it for figure 1

Figures 2, 3, 4 and 5: Please enlarged these figures to facilitate their reading.

Line 399: Precise if the same preparation of 2,3-dehydrosilybin described in ref 40 was used here. If not please add in short the structural characterization of this newly prepared compound.

Author Response

The authors of "Differential Effects of Flavonolignans Silybin, Silychristin and 2,3-Dehydrosilybin on Mesocestoides vogae Larvae (Cestoda) under Hypoxic and Aerobic in vitro Conditions" report on a very complete set of experiments providing a clear and sound view of the effects of 3 different flavonolignans on cestoda larvae under both aerobic and hypoxic conditions.

I strongly recommend this work for publication following (very) minor revisions.

We are grateful for positive review.

Line55 / Figure 1: silydianin is not shown. Change the text or draw it for figure 1

The structure of silydianin has been added to fig. 1. Figure caption was changed accordingly.

Figures 2, 3, 4 and 5: Please enlarged these figures to facilitate their reading.

The figures size conforms to standards in journals and output resolution of programs used. We provided new figure source files in high resolution to editorial office for use in publishing process.

Line 399: Precise if the same preparation of 2,3-dehydrosilybin described in ref 40 was used here. If not please add in short the structural characterization of this newly prepared compound.

The same preparation was used. The text was clarified.

Reviewer 2 Report

Dear Authors,

The manuscript titled ""Differential effects of flavonolignans silybin, silychristin and 2,3-dehydrosilybin on mesocestoides vogae larvae (cestoda) under hypoxic and aerobic in vitro conditions" is interesting and carefully written. Every aspect of the study is good explained, that is why I have no remarks.

Author Response

Dear Authors,

The manuscript titled ""Differential effects of flavonolignans silybin, silychristin and 2,3-dehydrosilybin on mesocestoides vogae larvae (cestoda) under hypoxic and aerobic in vitro conditions" is interesting and carefully written. Every aspect of the study is good explained, that is why I have no remarks.

We are very grateful for this review; there is always something to improve.